# Fitness consequences of targeted gene flow to counter impacts of drying climates on terrestrial-breeding frogs

Tabitha S. Rudin-Bitterli[1,2], Jonathan P. Evans[1,2] & Nicola J. Mitchell [1✉]

Targeted gene flow (TGF) could bolster the adaptive potential of isolated populations threatened by climate change, but could also lead to outbreeding depression. Here, we explore these possibilities by creating mixed- and within-population crosses in a terrestrial-breeding frog species threatened by a drying climate. We reared embryos of the crawling frog (*Pseudophryne guentheri*) on wet and dry soils and quantified fitness-related traits upon hatching. TGF produced mixed outcomes in hybrids, which depended on crossing direction (origin of gametes from each sex). North-south crosses led to low embryonic survival if eggs were of a southern origin, and high malformation rates when eggs were from a northern population. Conversely, east-west crosses led to one instance of hybrid vigour, evident by increased fitness and desiccation tolerance of hybrid offspring relative to offspring produced from within-population crosses. These contrasting results highlight the need to experimentally evaluate the outcomes of TGF for focal species across generations prior to implementing management actions.

[1] School of Biological Sciences, The University of Western Australia, Crawley, WA 6009, Australia. [2] Centre for Evolutionary Biology, The University of Western Australia, Crawley, WA 6009, Australia. ✉email: nicola.mitchell@uwa.edu.au

Genetic variation in traits related to environmental tolerance can allow populations to rapidly adapt to environmental change[1,2]. Yet, as habitats become fragmented, gene flow between isolated populations is constrained[3–5], reducing genetic diversity and adaptive capacity. To mitigate these problems, targeted gene flow (TGF, also known as assisted gene flow) is suggested for future management of threatened plant[6–8] and animal species[9–11]. TGF involves the deliberate translocation of individuals or gametes within a species' indigenous range to increase the genetic variation (and concurrently the fitness) of recipient populations, and to promote their adaptation to anticipated local conditions[10,12]. Many species show geographically based variation in adaptive traits[13–15], and therefore TGF can potentially increase genetic variation of recipient populations in a desired direction[16]. In this way, TGF can potentially reinstate or increase natural gene flow between populations to facilitate adaptation[16,17].

The current enthusiasm for TGF is tempered by concerns over risks associated with mixing previously disjunct populations and subsequent outbreeding depression, where offspring of local and introduced parents have a lower fitness than pure-cross offspring of the recipient population[18–21]. Outbreeding depression can arise in F1 or later generations as a result of genetic incompatibilities (e.g., via underdominance or epistatic interactions[22]) or via the dilution of local adaptation in the recipient population (e.g., if populations are adapted to conditions other than climate[23]). While the risks of outbreeding depression may be overemphasised for some taxa[10,12,20,24], the consequences of mixing populations are difficult to predict[18,25]. It is therefore important to assess the potential risks and benefits of TGF empirically, something only a handful of studies have done to date[11,16,26–30].

Amphibians are among the most vulnerable taxa to climate change[31,32]. Many species have undergone climate-related extinctions or declines[32–35], and species distribution models suggest that a further 12–47% of anuran species are at risk of extinction due to climate change[36]. Reduced precipitation is an obvious threat, as unshelled eggs and highly permeable skin render all amphibian life stages susceptible to desiccation and subsequent mortality[33]. However, as some amphibian populations show greater tolerance to desiccation stress than others[37], TGF could be strategically employed to enhance the resilience of populations adapted to higher-rainfall regions that are now drying.

Here, we explore the potential of TGF to mitigate declines in population-level fitness using the crawling frog, *Pseudophryne guentheri*, a species in which populations are threatened by habitat loss and declining winter rainfall (see *Methods*). Application of TGF could potentially mitigate these dual threats to population persistence, but as outbreeding depression has been shown in laboratory crosses of a congeneric species[30], TGF could also be ineffective, or have deleterious consequences. We evaluated TGF within a laboratory setting by creating pure and reciprocal crosses among four geographically distant populations. We then assessed phenotypic traits in the resulting offspring, comparing individuals reared on wet soils (−10 kPa, a benign treatment) to those reared on drier soils (−400 kPa) that significantly reduce survival and hatchling fitness, particularly in mesic populations[37,38]. Our experiments were designed to answer three questions: (1) do offspring from between-population crosses show hybrid vigour or reduced fitness; (2) does the effect depend on geographic or genetic distance; and (3) does the direction of the cross influence TGF outcomes?

## Results

We sourced *P. guentheri* from low-rainfall regions at the northern edge of the species' range and from higher-rainfall regions close to the centre the species range (Fig. 1 and Table 1). We then used linear mixed-effects models to compare offspring traits from pure and hybrid crosses reared under two environments. We focused on a range of traits putatively tied to fitness, including embryonic survival, the time required to hatch after inundation, the proportion of malformed hatchlings, and wet weight, developmental stage and swimming performance at hatching. Across all within- and between-population crosses created in this experiment, low soil moisture significantly reduced embryonic survival, increased the time required to hatch after inundation, reduced the wet weight and developmental stage of hatchlings, and increased the proportion of malformed hatchlings (Supplementary Table 1 and Fig. 2). Soil moisture also affected the swimming performance of hatchlings, with swimming velocity decreasing in hatchlings reared on dry soils (Supplementary Table 2 and Fig. 3).

Fitness-related traits differed remarkably among crosses due to a complex interplay between many of the fixed effects we considered (soil moisture, female population origin, male population origin, and all interactions). The population origin of females was a significant effect in six of the nine traits examined (Supplementary

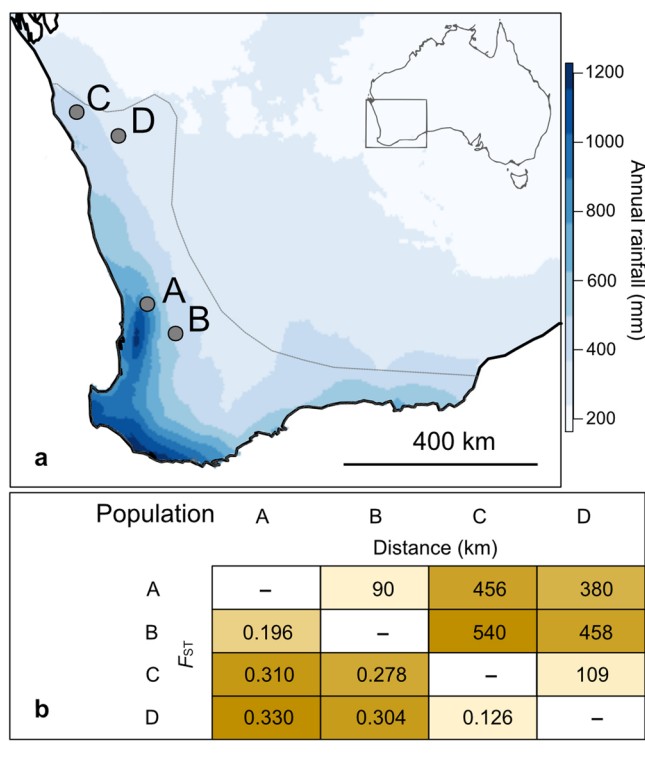

| Population | A | B | C | D |
|---|---|---|---|---|
| | Distance (km) | | | |
| A | – | 90 | 456 | 380 |
| B | 0.196 | – | 540 | 458 |
| C | 0.310 | 0.278 | – | 109 |
| D | 0.330 | 0.304 | 0.126 | – |

*(F_ST values in lower-left triangle; Distance (km) in upper-right triangle)*

**Fig. 1 Details of four breeding populations in southwestern Australia from which adult *P. guentheri* were collected. a** The species' distribution (grey line; based on occurrence records from the Atlas of Living Australia) spans a ~300–1250 mm annual rainfall gradient, and both adults and terrestrial embryonic stages show clinal variation in desiccation tolerance, with range edge populations being more resistant to dry conditions than populations at the core of the species range[37]. **b** Matrix of pairwise geographic and genetic distances between populations, where genetic distances ($F_{ST}$) were calculated in an earlier study[39] using 12,787 selectively neutral SNP loci. **c** a gravid female (collected from all sites except D). **d** an adult male (collected from all sites).

**Table 1 Site characteristics, sample numbers and genetic diversity estimates[39] ($H_e$ = mean expected heterozygosity, $F_{IS}$ = inbreeding coefficient) for each *P. guentheri* population.**

| Population name (code) | Latitude | Longitude | N | | | Annual mean precipitation (mm)[a] | $H_e$ | $F_{IS}$ |
|---|---|---|---|---|---|---|---|---|
| | | | ♂ | ♀ | $F_1$ | | | |
| Chidlow (A) | 31°53′05.5″S | 116°18′48.0″E | 15 | 5 | 622 | 788 | 0.217 | 0.304 |
| Ridgefield (B) | 32°28′25.9″S | 116°58′27.9″E | 16 | 13 | 1548 | 428 | 0.226 | 0.342 |
| Binnu (C) | 28°02′30.8″S | 114°39′36.0″E | 15 | 6 | 803 | 352 | 0.215 | 0.281 |
| Mullewa (D) | 28°31′07.3″S | 115°38′11.4″E | 15 | - | - | 329 | 0.193 | 0.220 |

[a]Rainfall data are interpolated average values (1980–2017)[40] for the coordinates of each population.

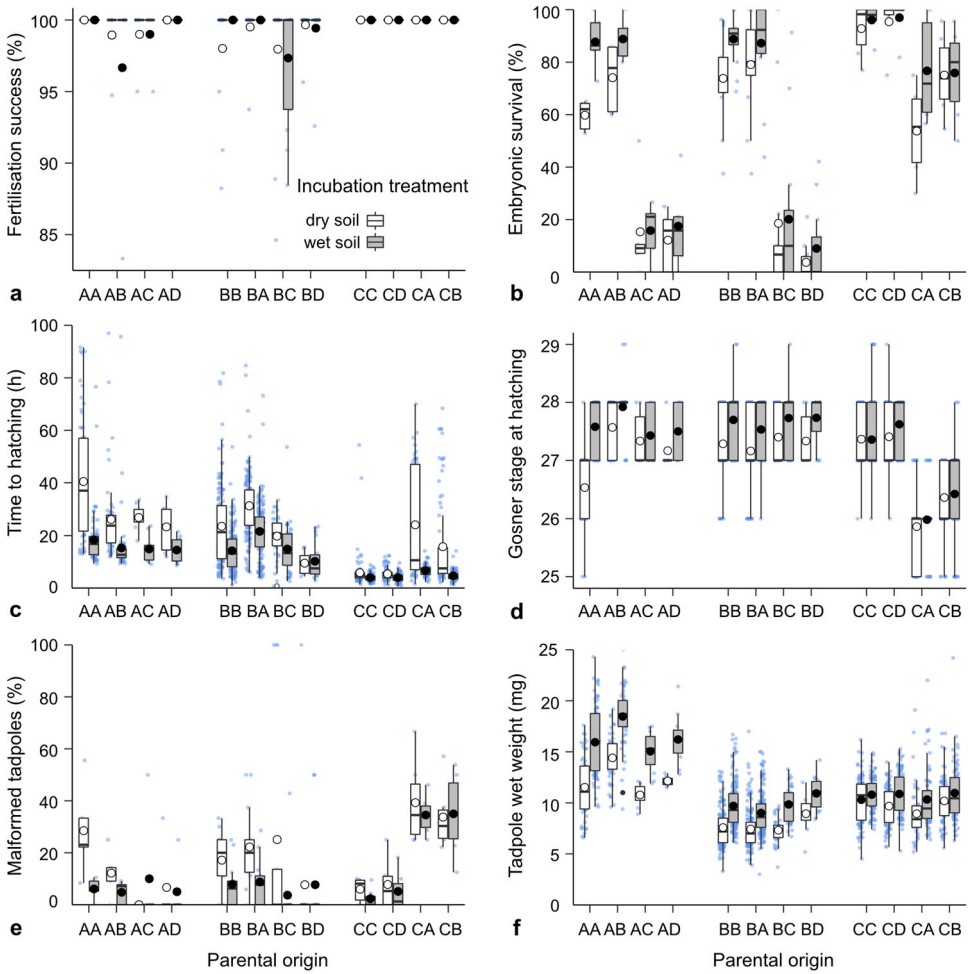

**Fig. 2 Fertilisation success and embryonic and hatchling trait responses to dry and wet rearing environments in within- and between-population crosses of four populations.** Fertilisation success was measured for 92 population crosses (**a**) and traits were measured for between 1460 and 2793 embryos and tadpoles (**b–f**, see Supplementary Table 1). The first letter of the parental origin represents the female (population) origin, and the second letter represents the male origin. Mean responses (large white or black circles, according to treatment) are superimposed over boxplots, which are superimposed over raw data (jittered small blue circles). Lower and upper boxplot boundaries are 25th and 75th percentiles, respectively, the thicker line inside the box is the median value, and lower and upper error bars are 10th and 90th percentile, respectively.

Tables 1 and 2). For example, hatching wet mass was greatest in offspring from females from population A, and lowest in offspring where females originated from populations B and C, irrespective of ovum size. Male origin effects were also strong and significantly affected all but one offspring trait (Supplementary Tables 1 and 2), and female origin-by-male origin interactions were highly significant in all traits (Supplementary Tables 1 and 2). This latter result showed that the interaction between female and male gametes affects offspring fitness in hybrid and non-hybrid crosses, and that the direction of the cross (e.g., AB versus BA, where the first letter indicates the origin of the egg) influences the outcome. Additionally, many of the female, male, and female-by-male origin effects were modified by the developmental environment (i.e., wet or dry soils; Supplementary Tables 1 and 2). In light of these complex patterns, the outcomes of each type of cross are discussed in turn below.

**Within-population crosses (AA, BB & CC).** Within-population (pure) crosses differed in their desiccation tolerance in a pattern

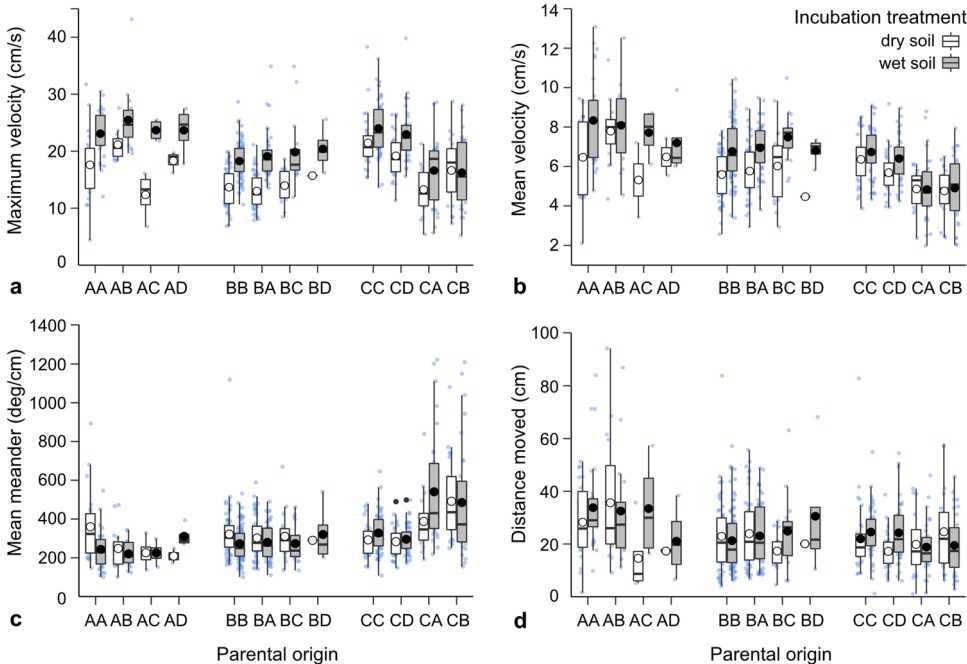

**Fig. 3 Swimming performance traits (a–d) of hatchlings reared at two soil treatments (dry and wet) in within- and between-population crosses of four populations.** Traits were measured for 633 tadpoles (Supplementary Table 2). The first letter of the parental origin represents the female (population) origin, and the second letter represents the male origin. Mean responses (large white or black circles, according to treatment) are superimposed over boxplots, which are superimposed over raw data (jittered small blue circles). Lower and upper boxplot boundaries are 25th and 75th percentiles, respectively, the thicker line inside the box is the median value, and lower and upper error bars are 10th and 90th percentiles, respectively.

consistent with site water availability, in agreement with a related study[37]. Offspring from pure crosses of the wettest population (AA) were most sensitive to dry conditions, with embryonic survival, hatchling wet weight, and developmental stage decreasing markedly, and hatchling malformations becoming more prominent if embryos were reared on dry soils (Fig. 2). Furthermore, swimming performance of AA hatchlings reared on dry soils was reduced substantially, with hatchlings swimming more slowly and less linearly than those from the wetter rearing environment (Fig. 3). In contrast, offspring from pure crosses of population C, situated near the northern range edge, showed little response to the dry rearing treatment (CC in Figs. 2 and 3).

**Long-distance (north-south) hybrid crosses (AC, AD, BC, BD, CA, CB).** Embryonic survival was very low (~15%) when eggs from southern populations A or B were crossed with sperm from northern populations C or D, both in the dry and wet soil treatments (Fig. 2a). Offspring from these crosses that did survive to hatching, however, hatched more quickly in the dry treatment compared to their pure-cross counterparts (Fig. 2b). Furthermore, the developmental stage at hatching in AC and AD crosses was more advanced (average Gosner stage 27.3) in the dry treatment compared to pure AA crosses (average Gosner stage 26.5), indicating slightly more rapid development (Fig. 2d).

Offspring originating from long-distance crosses in the opposite direction (eggs from northern population C crossed with sperm from southern populations A or B), also had reduced survival in dry and wet treatments compared to pure CC crosses, although these effects were much less severe, with at least 55% of embryos hatching (Fig. 2a). Approximately one-third of hatchlings from these crosses were malformed, irrespective of the rearing environment (Fig. 2e), and their swimming performance was poor (Fig. 3). Furthermore, the developmental stage at hatching was lower compared to CC crosses (Fig. 2d) and the

time required to hatch after inundation was greater in the dry treatment (Fig. 2b).

**Shorter-distance (east-west) hybrid crosses (AB, BA, CD).** Offspring originating from hybrid crosses of two populations located near the centre of the species' range (AB; eggs from population A crossed with sperm from population B) showed enhanced desiccation tolerance compared to AA crosses in all traits studied. As such, embryonic survival increased, and the time required to hatch and the number of hatchling malformations decreased in AB offspring reared on dry soils (Fig. 2a, b, d). Additionally, tadpole wet mass and developmental stage at hatching were greater in AB offspring compared to either pure-cross counterparts (AA or BB; Fig. 2c, d), both in the wet and dry treatments, and their swimming performance was enhanced (Fig. 3).

Offspring originating from short-distance crosses in the opposite direction (eggs from population B crossed with sperm from population A) took longer to hatch after inundation relative to BB crosses, and their developmental stage at hatching was reduced slightly, both in the wet and dry treatments (Fig. 2b, d). Otherwise, BA crosses had similar fitness to pure BB crosses.

Offspring originating from short-distance crosses between the two populations at the northern edge the species' range (CD; eggs of population C crossed with sperm from population D) barely differed in desiccation tolerance and fitness traits compared to CC offspring. Wet mass at hatching was increased slightly in the dry treatment, and developmental stage at hatching was marginally higher in the wet treatment in CD crosses, compared to pure-cross counterparts (CC).

## Discussion
Reciprocal crosses between three of four *P. guentheri* populations resulted in exceptionally diverse outcomes for traits putatively linked to fitness (Fig. 4). Reduced fitness, potentially attributable to outbreeding depression, was apparent in all offspring

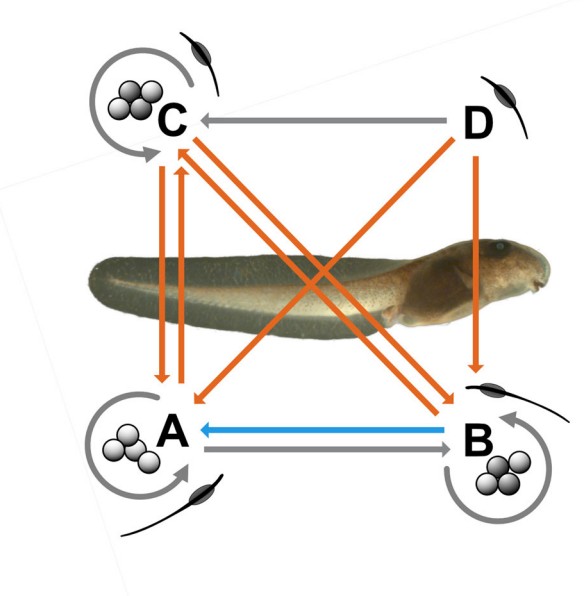

**Fig. 4 Summary of the effects of male (population) origin on fitness-related traits in *P. guentheri* hatchlings, where orange arrows denote negative effects, blue arrows positive effects and grey arrows neutral effects.** Arrow directions indicate the mixing of sperm with eggs in populations A, B and C. As we did not obtain gravid females from population D, only crosses between populations A, B and C were reciprocal. The background image shows an example of a BB hatchling, and sperm images indicate different sperm morphology between A/B and C/D males.

originating from long-distance crosses (north-south), whereas shorter-distance crosses (east-west) led to two neutral outcomes and one instance of hybrid vigour across all traits measured. Intriguingly, the origin of the gametes greatly influenced offspring traits in the one east-west reciprocal cross; sperm from population B mixed with eggs from population A produced desiccation-tolerant phenotypes, but not vice versa. Asymmetries were also apparent in north-south crosses, where very low embryonic survival resulted from northern sperm/southern eggs, but better survival yet high malformation rates in hatchlings produced from southern sperm and northern eggs. While lending support to the view that the risk of outbreeding depression increases with genetic, geographic and environmental distance[21], the fittest progeny were produced by mixing two genetically distinct populations, physically separated by 100 km, and this fitness advantage (relative to pure crosses) was achieved both under benign developmental environments, and under more stressful conditions that may simulate future environments. This result is in stark contrast to outbreeding depression inferred in a related species (*P. bibronii*), where populations just 20 km apart were mixed and produced largely inviable offspring[30]. We discuss our key findings in turn below, and conclude with commentary on the potential application of TGF in this genus.

Our demonstration of reduced fitness in F1 hybrids originating from long-distance (~460 km) north-south crosses is consistent with asymmetric reproductive isolation, as reciprocal crosses produced different levels of fertilisation success and hybrid inviability. Asymmetric reproductive isolation is common at the species level[41] and is frequently the result of Dobzhansky-Muller incompatibilities, where two species accumulate complementary genes that have no deleterious effects within species, but cause

inviability or sterility when combined with genes from another species (complementary epistasis between loci)[41–44]. Additionally, cytonuclear incompatibility (=discordance) is likely to explain asymmetrical outcomes in reciprocal crosses of two species of newt (*Triturus cristatus* and *T. marmoratus*)[45], and differential growth performance across mitotypes of the salamanders *Ambystoma macrodactylum*[46]. In our current study, reproductive isolation is positively correlated with pairwise genetic distance (and thus, presumably, time of divergence)[47], and it appears that the drier range edge (northern) and mesic (southern) populations of *P. guentheri* have sufficient genetic divergence for reproductive isolation to occur.

In frogs, the lower threshold for the evolution of hybrid inviability is estimated to be a Nei's genetic distance ($D_A$) of 0.3[48]. Genetic analysis of 12 *P. guentheri* populations, based on >12,000 single-nucleotide polymorphism (SNP) markers, revealed that the genetic distance between the northern (C, D) and southern (A, B) populations used in this study was well below this threshold ($D_A = 0.14$)[39]. However, strong sexual selection might facilitate rapid evolution of reproductive isolation, even when the genetic distance between populations is low[49]. Furthermore, the relationship between hybrid inviability and genetic divergence can differ greatly, even between closely related taxa[18]. For example, crosses between *Drosophila* species that diverged 0.35 Myr ago resulted in outbreeding depression in the F2[50], while hybridisations between other *Drosophila* species isolated for much longer (3–4.3 Myr[51]) resulted in F2 heterosis[52]. This makes it difficult to predict the severity of outbreeding depression in advance, and builds the case to experimentally evaluate the outcomes of TGF for focal species across generations[16].

Recent findings for *P. guentheri* have revealed considerable differences in sperm characteristics between the northern and southern populations used in this study[53]. For example, spermatozoa from populations C and D swum more slowly and in a less linear fashion than sperm from populations A and B, and a smaller proportion of C/D sperm were active[53]. Furthermore, C/D spermatozoa were smaller (both shorter tails and heads) with broader heads[53]. Eggs from populations A and B also took substantially longer to show signs of fertilisation when mixed with sperm from populations C and D (see *Methods*). Therefore, atypical spermatozoa–egg interactions[54] may have contributed to slow fertilisation rates in long-distance crosses, highlighting that non-genetic constraints may affect the application of TGF in conservation programmes.

Despite forming distinct genetic clusters[39], short-distance (~100 km) crosses between populations A and B led to asymmetric hybrid vigour. When sperm from population B fertilised eggs from population A (the most mesic site in this study), the resulting offspring showed enhanced desiccation tolerance relative to AA offspring. These effects were very strong in most traits measured, to the extent that trait responses of AB hatchlings resembled those of BB offspring in the drier treatment. For example, embryonic survival of AB hybrids was greater (22% difference), time to hatching was faster (37% difference) and the number of hatchling malformations was reduced (46% difference) in the drier treatment relative to AA offspring (Fig. 2). Furthermore, hatchling wet mass and developmental stage in hybrids were greater compared to their AA counterparts, and their swimming performance was enhanced (Figs. 2 and 3).

These trait differences are likely to provide fitness benefits in a drier climate. For example, if hatching-stage embryos breach their egg capsules more quickly when flooded, the resulting larvae are more likely to be washed into pools of standing water where they can feed and seek refuge. Furthermore, faster development rates allow embryos to hatch earlier if an appropriate stimulus is received (flooding lowers $PO_2$ and thereby triggers hatching in a

related species[55]), which lowers risks such as predation or fungal infestation of eggs[56,57]. Finally, malformations and body size affect swimming performance at hatching, which in turn influences predator avoidance and foraging efficiency[58–63]. As malformations can persist past metamorphosis[64], they are likely to have long-term consequences on survival and fitness.

Whether the increased desiccation tolerance in AB offspring is the result of hybrid vigour, introduction of desiccation-tolerant genes, or both, is difficult to disentangle. Importantly, the cause of these effects matters greatly for the long-term outcomes of TGF. Hybrid vigour most commonly arises due to immigrant alleles masking deleterious alleles in the receiving population[65,66]. Any increase in population fitness following hybridisation is expressed particularly strongly in the F1 generation, but is likely to disappear in subsequent generations when potentially deleterious interactions among recessive alleles at different loci become exposed to selection due to recombination[67]. There are numerous examples of outbreeding depression following initial heterosis across a range of taxa, including invertebrates[19,68], birds[69] and mammals[70]. Yet, if enhanced performance of AB hybrids is the result of introduction of desiccation-tolerant genes from population B, then the positive effects observed here are likely to persist in the long-term, and selection may favour the establishment of these genes within the recipient population.

Although multi-generational studies are needed to separate the origin of these effects, comparison of trait responses between pure and hybrid crosses suggests that introduction of pre-adapted genes may have contributed to faster hatching (and at a more advanced developmental stage) in BA hybrids in the dry treatment. In pure crosses, both traits varied among populations in a pattern consistent with local water availability, with AA offspring (from the most mesic population) being the slowest to hatch, and at a less-advanced developmental stage, and CC offspring (from the driest population where we obtained females) hatching and developing the fastest in the drier treatment. If males carry locally adapted genes for these traits, we would expect their expression to be altered in hybrids in a pattern that reflects annual rainfall (or related abiotic variables) at the sites from which males were sourced. Accordingly, AB, AC and AD hybrids should hatch more quickly in the drier treatment compared to AA offspring. Analogously, BA offspring would be expected to take longer to hatch in the drier treatment than BB offspring (whereas BC and BD offspring should hatch more quickly), and CA and CB offspring should take longer to hatch compared to CC offspring. Our data (Fig. 2b, d) show this pattern for the 'time to hatching' trait, and for developmental stage at hatching.

The strength of male population origin effects, which exceeded treatment effects in both traits (Supplementary Table 1), further underscores the role that male population origin plays in altering the desiccation tolerance of hybrids. Alternatively, the negative effects of genetic incompatibility may be exacerbated under stressful environments[71], consequently leading to different offspring fitness and desiccation tolerance amongst hybrid crosses. However, given that positive outcomes were observed in some crosses (e.g., the time to hatching was reduced), despite genetic incompatibilities (low embryonic survival), this latter explanation is less likely. Another possibility is that locally adapted phenotypes produced in different environments reflect interactions between mitochondrial genes (transmitted via females) and nuclear genes, and so mitotypes might modulate the expression of traits passed on by males. Mitonuclear/environmental interactions have recently been shown in insects[72] and birds[73], but mitonuclear co-evolution has received limited attention in amphibian populations[46]. In contrast, polyploidy is well documented in anuran amphibians[74], and dosage effects in polyploid lineages could also potentially explain asymmetry in phenotypes

created by crosses of the same populations in reciprocal directions. As polyploidy may be generated via hybridisation[74], future research using this model system could investigate chromosome numbers in parents and offspring to refute or confirm the possibility that polyploids exist or can be created in *P. guentheri*.

The ideas presented above are necessarily speculative, and we are a long way from understanding how admixture of populations will play out in the short and longer term. An encouraging result in our focal system is that desired traits (in this instance those related to desiccation tolerance) can be achieved by mixing male gametes from the 'pre-adapted' population with female gametes in the target population. This finding has great practical importance; females are generally difficult to locate, and may ovulate and sometimes expel eggs in transit, so it would be logistically challenging to move their gametes into a target population at scale. In contrast, calling males can be easily collected from a breeding chorus, and their gametes harvested and stored for up to two weeks[75]. Hence in practice, multi-male sperm suspensions (as used here—see *Methods*) could be pipetted onto egg masses freshly stripped from females within a field setting, effectively seeding a population with new genetic material if most females can be intercepted before mating. Indeed, it has been shown in some *Pseudophryne* populations that almost all females entering a breeding chorus can be captured[76], so the logistics of achieving TGF in this species are tractable, and less complex than those for internally fertilised amniotes (reptiles, birds and mammals).

There is growing interest in Australia for exploring TGF for threatened species[16], as well as investment in genomic resources that could aid in targeting adaptive traits[77]. Our results demonstrate that the outcomes of TGF in *P. guentheri* depend on the particular combination of source and recipient population, although surprisingly in one reciprocal cross this depended on the origin of the specific gametes. Further work is needed to clarify whether this result was due to the introduction of pre-adapted alleles, heterosis, mitochondrial/genomic interactions, polyploidy, or as yet unrecognised processes. However, the strong hybrid vigour and increased desiccation tolerance in AB crosses is promising, particularly considering that the parental populations are genetically divergent ($F_{ST}$ 0.196).

Notably, toadlets in the *Pseudophryne* genus include Critically Endangered species (e.g., northern and southern Corroboree frogs), where genetic augmentation to improve resistance to the chyrid fungus (i.e., a form of TGF), is being considered[30,78]. Coincidentally, these species were crossed in a genetic study in the late 1980s[79], and hybrids (i.e., *P. pengilleyi* x *P. corroboree*) had high embryonic survival (~85%)—as did the more successful crosses in our study—and better survival to hatching and metamorphosis than intra-population crosses, suggestive of genetic rescue[26]. These and several other threatened amphibian species are subject to captive breeding programmes and efforts to improve the capacity to generate, harvest and store gametes[80,81]. Given that application of TGF in these species will require both a reliable source of adults, and a supply of viable eggs and sperm, we are well positioned to initiate laboratory trials that target the expression of traits that could enhance survival of wild populations as climates become increasingly stressful.

Our results also highlight risks of TGF, as evident by apparent outbreeding depression in some long-distance crosses at levels that suggest reproductive isolation. As the effects we observed depended on the direction of the cross, we expect the outcomes of TGF will be difficult to predict theoretically. This point is especially pertinent considering that strong outbreeding depression has been shown in population crosses of a congener (*P. bibronii*)[30], while in another congener (*P. coriacea*) the observation that females preferentially mate with related males was explained as a mechanism to avoid outbreeding depression[82].

Consequently, as highlighted in a recent review[15], empirical tests of TGF in model species (as was demonstrated here), and the monitoring of long-term consequences, are essential. Otherwise, managers have few tools at their disposal to enhance the adaptive potential of populations and species threatened by climate change.

## Methods

**Ethics.** All animal procedures were approved by the University of Western Australia's (UWA) Animal Ethics Committee (permit number RA/3/100/1510), and the research was conducted under license 08-000560-1 from the Western Australian Department of Biodiversity, Conservation and Attractions.

**Study species and population collections.** Female *Pseudophryne guentheri* deposit large clutches of eggs that are fertilised externally, and embryos develop in terrestrial locations (burrows, depressions under cover) until suspending development at Gosner[83] stages 26–28[38,82]. Larvae are cued to hatch by flooding, and develop aquatically thereafter, reaching metamorphosis in around three months[84]. The winter rainfalls that promote breeding in this species have declined markedly in recent decades (19% reduction since the 1970s)[85–87], and are projected to further decline by up to 30% by 2090[85]. Additionally, many *P. guentheri* populations are now isolated due to extensive clearing of habitat[88,89], and significant declines in genetic diversity (expected heterozygosity and allelic richness) associated with repeated drought have been revealed through genomic analysis[39].

Reliable in vitro fertilisation techniques developed for this species[38,90] provided an opportunity to employ a highly tractable framework to examine the outcomes of TGF. We collected adult *P. guentheri* from four geographically separated breeding sites, situated at two latitudes, in May and June 2017 (Table 1). Sites spanned a ~460 mm annual rainfall gradient, with site A receiving the most rain per year and site D receiving the least (Table 1 and Fig. 1). *Pseudophryne guentheri* collected from breeding populations at each site show variation in desiccation tolerance, with adults and embryos from site A being the most sensitive to dry conditions[37]. Population genetic analysis[39] has demonstrated high levels of inbreeding in all populations (Table 1), and genetic differentiation among *P. guentheri* populations is high (overall $F_{ST} = 0.186$), which suggests low levels of contemporary dispersal. *P. guentheri* from sites A and B form distinct genetic clusters[39], indicating low historical gene flow despite their close proximity (100 km), whereas *P. guentheri* from sites C and D show admixture, but are genetically distinct from populations A and B[39].

In total, 15–16 calling males from each population were collected by hand and in pit-fall traps. Gravid females were more difficult to collect due to their cryptic behaviours, and so sampling was restricted to 5–13 females from each of three sites (A, B and C; Table 1). All frogs were temporarily housed in small (4.4 L) plastic terraria containing moist sphagnum moss, and transported to the University of Western Australia within two days of collection. There, frogs were fed a diet of pinhead crickets and kept in a controlled-temperature room at 16 °C with an 11/13 h light/dark photoperiod to mimic winter conditions.

**Breeding design and in vitro fertilisations.** Egg clutches of each female were divided equally into four groups, and fertilised with sperm from males originating from each of the four populations, resulting in one pure and three hybrid crosses. To control for potential parental compatibility (i.e., specific pairwise male-by-female) effects on offspring fitness[38,71], a sperm mixture, containing sperm from five random males from the appropriate population, was used to fertilise the eggs of each female in each population[91].

Sperm was obtained from testes macerates after euthanizing males via ventral immersion in < 0.03% benzocaine solution, followed by double pithing. Sperm was stored on ice in 25–458 μL (adjusted according to the weight of the testes) standard amphibian ringer (SAR; 113 mM NaCl, 2 mM KCl, 1.35 mM CaCl₂). This buffer allows storage of sperm for extended periods (days–weeks) without substantial declines in motility[92,93]. Sperm concentrations were measured using an improved Neubauer haemocytometer (Hirschmann Laborgeräte, Eberstadt, Germany) and sperm suspensions were diluted with 1:1 SAR to 100 sperm per μL.

Upon arrival at the laboratory, females were gently squeezed to determine whether ovulation had occurred. Approximately 35% of females had ovulated naturally while in transit and their eggs were gently stripped. For the remaining females, ovulation was induced via two subcutaneous injections of the hormone LHRHa over the course of two days[37,90]. Approximately 10 h after the second injection, eggs were gently stripped from each female. In all instances, freshly stripped eggs were moistened with SAR and distributed equally among four small petri dishes. A standardised number of sperm from five random males collected at each site was pipetted onto one edge of each petri dish, mixed gently with the pipette tip, and then activated with a pre-calculated volume of 1:4 SAR solution[38]. This resulted in eggs from all females in the experiment being fertilised from males collected from sites A, B, C and D. Each dish was then manually agitated for 20 s to promote fertilisation. After 15 min, eggs were temporarily submerged in water, backlit and photographed using a digital imaging camera (Leica DFC320) attached to a light microscope (Leica MZ7.5) at 6.3 X magnification. These images were used

to measure the ovum diameter of 50 randomly selected eggs from each female, using ImageJ software[94]. Fertilisation success was initially scored one h after mixing eggs and sperm by counting eggs that had rotated (Gosner[83] Stage 1). However, eggs from populations A and B took substantially longer to show signs of fertilisation when mixed with sperm from populations C and D. We therefore scored fertilisation success a second time, six hours after sperm and eggs were mixed.

**Incubation treatments.** Fertilised eggs from each cross were reared on sandy loam soil at two water potentials (ψ): a wet soil (ψ = −10 kPa) and dry soil (ψ = −400 kPa). The soil was previously collected from a separate *P. guentheri* breeding site, and the soil water potentials represented a range found in natural nest sites[38]. Embryo incubation and soil preparation were performed as described in Rudin-Bitterli et al.[37]. Briefly, soil was oven-dried at 80 °C for 24 h, distributed into small containers and rewetted with an appropriate mass of deionised water using a water-retention curve previously determined for the soil sample. The water content of the soil (g/g/ of oven dry soil) was approximately 50% in the wet treatment, and 21% in the dry treatment, and containers were sealed with a lid after wetting. Fertilised eggs from each cross were selected at random and distributed onto soils within 7–9 h of fertilisation. Small plastic rings (nylon plumbing olives, 12 mm in diameter) were labelled and placed around eggs to identify individual crosses. Sealed containers were then placed in incubators set at 16 ± 0.5 °C, and embryos were monitored every two days. Any dead eggs were removed and discarded.

**Response variables.** Putative fitness from within- and between-population crosses, reared in dry and wet rearing environments, was assessed at hatching. At 33 days after fertilisation (when embryos were approximately at Gosner[83] Stage 26), hatching was induced by placing embryos individually in small test tubes containing 2 mL of deionised water[38]. Embryos were then monitored at least every 30 min until hatching, defined as when an individual completely escaped their egg capsule. Embryonic survival was recorded for each family as the percentage of fertilised eggs that hatched.

Swimming performance was recorded 6–12 h after hatching on a subset of hatchlings ($N = 633$ across all within and between-population crosses). For this purpose, individual hatchlings were placed in a petri dish (diameter = 150 mm) containing water 10 mm deep. After an initial acclimation period of 1 min, the tail of each hatchling was nudged with a glass cannula to elicit a burst swimming response. A video camera (Canon PowerShot G16, recording at 60 fps) installed 300 mm above the petri dish was used to film three burst swimming responses for each hatchling, and their movement was later tracked and analysed using EthoVision v8.5 software[95]. EthoVision enabled the quantification of the following swimming parameters: maximum velocity (cm s⁻¹), mean velocity (cm s⁻¹) and total distance moved (cm). We also recorded mean meander (deg cm⁻¹), a measure of the straightness of the swimming response, as dry rearing environments can lead to asymmetrically shaped hatchlings[37,38] that swim in a more circular motion. A hatchling was considered to be moving when it exceeded 0.45 cm s⁻¹. As each video recording contained three burst swimming responses with periods of no movement in between them, EthoVision only analysed frames in which a hatchling moved faster than 0.45 cm s⁻¹ (consequently merging the three swimming responses for each hatchling). Immediately following the swimming performance trials, hatchlings were euthanized in <0.03% benzocaine and preserved in 10% neutral buffered formalin.

Wet masses of preserved hatchlings were recorded to the nearest 0.001 g after blotting on tissue. Hatchlings were then photographed in lateral view (while submerged in water to minimise refraction) using a digital imaging camera (Leica DFC320) attached to a light microscope (Leica MZ7.5) at x 6.3 magnification. These images were used to score malformations for each hatchling and to determine their developmental stage[83] by examining the hind limb buds.

**Statistics and reproducibility.** For each trait measured, sample sizes varied between 630 and 2973 offspring created from pure and mixed sources (Tables S1, S2). All analyses were performed in R version 3.4.3 (R Development Core Team 2017). Linear mixed-effects models (with restricted maximum-likelihood methods; REML) were run using the lme4 package[96] to compare offspring traits from pure and hybrid crosses. In these models, treatment, female (population) origin, male (population) origin and all interactions (female origin-by-male origin, female origin-by-treatment, male origin-by-treatment, female origin-by-male origin-by-treatment) were considered as fixed factors. We checked for overdispersion in our models using the 'overdisp_fun' function proposed by Bolker et al.[97]. Only one trait, embryonic survival, was overdispersed, which if uncontrolled can lead to biased parameter estimates. To account for this overdispersion we included an observation-level random effect (where each data point in our analysis receives a unique level of random effect) when analysing this trait[98]. As the eggs from each female were subjected to a split-clutch design, the term for dam (i.e., individual female ID) was added as a random effect to account for the use of individual females across multiple fertilisation events. We also included ovum size as a covariate in all analyses to control for possible maternal effects arising from different

patterns of egg provisioning among females[38]. The significance of the fixed effects was evaluated using Wald chi-squared tests.

Embryonic survival and hatchling malformation data were binomial variables and thus a generalised-linear mixed-effects model (GLMM) with a logit-link function was used for the analysis of these traits. In these models we included treatment, female (population) origin, male (population) origin and all interactions (see above) as fixed effects. As above, dam ID was treated as a random effect and ovum size was added as a covariate to control for possible maternal effects[38]. The significance of the fixed effects was evaluated using Wald Z-tests.

**Reporting summary**. Further information on research design is available in the Nature Research Reporting Summary linked to this article.

## Data availability
Data files[99] supporting the findings of this study are available in Dryad with the identifier https://doi.org/10.5061/dryad.6m905qg09.

## Code availability
R code[99] used to analyse data are available in Dryad with the identifier https://doi.org/10.5061/dryad.6m905qg09.

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

## Acknowledgements

We thank Brighton Downing for assistance in the field and laboratory, and Deanne Cummins, Blair Bentley, Jon-Paul Emery and Emily Hoffmann for their help collecting adults from breeding choruses. Corne van der Linden provided the *P. guentheri* images in Fig. 1. This research was funded by the ANZ Holsworth Wildlife Research Endowment, the Australian Government's National Environmental Science Programme through the Threatened Species Recovery Hub and the School of Biological Sciences at the University of Western Australia, T. Rudin-Bitterli was supported by the International Postgraduate Research Scholarship and the C.F.H. & E.A. Jenkins Postgraduate Research Scholarship.

## Author contributions

T.R.-B., N.M. and J.E. conceived the study. T.R.-B. carried out the fieldwork and laboratory experiments, analysed the data, and wrote the first draft of the manuscript. N.M. produced the figures. All authors contributed to writing and revising the manuscript.

## Competing interests

The authors declare no competing interests.
