## [Peer Review File · Communications Biology]

Reviewers' comments:

Reviewer #1 (Remarks to the Author):

This is an interesting manuscript which presents new and valuable data on the fitness of frog hatchlings which were created through artificial insemination and reared at two different levels of water availability, comparing four populations and their hybrid combinations including reciprocal male/female crosses. The main incentive of these experiments is to mimic targeted gene flow, which could serve as a future conservation mitigation measure for a study species which is threatened by decreasing precipitation levels. As a main finding, the authors show that the experimental crosses markedly differ in a multitude of fitness-related traits (probably most strongly in embryo survival), which is broadly linked to the level of population differentiation but also difficult to predict in detail.

The study species is highly suited to address the study question, and a lot of attention to detail has gone into the measurements of fitness-related traits. The findings are clearly presented, and the conclusions are warranted by the presented data. The main drawback of the work is, however, that I somewhat fail to find an overarching take-home message. From a conservation perspective, the experimental findings did not lead to the development of a tangible management plan. From a conceptual perspective, the underlying reasons for the observed fitness responses could not be further investigated, mainly because the study individuals were not genotyped.

It might be a little cruel to directly quote the authors in this context, but the last sentence of the first, main summary Discussion paragraph highlights my main concern. The scope of given work is somewhat limited when ".....the potential application of TGF in this genus" (line 177) is listed as a main point of consideration, and not for example broader insights for TGFs in general. I am aware that the authors also digest their "key findings" (line 176), but I found the entire Discussion -as the authors themselves admit at line 288- at times rather speculative, and also a little unfocused throughout.

The manuscript could become vastly strengthened for example if the findings from the experimental crosses are underpinned by individual-specific genetic data – I wonder if the SNP genotypes published in a previous paper by the authors were based on the study individuals of the present manuscript? As one inference, the parentage of the hatchling cohorts could be determined (the mixing of sperm from 5 males leaves between 1 and 5 possible sires), to reveal possible effects of population crosses on levels of sperm competition. As a further possible inference, mitochondrial genotypes could be linked to the outcome of crosses, particularly in light of the finding that sex-reciprocal crosses between given population pairs lead to differential phenotypic fitness. At present, the authors largely neglect the possible presence of cytonuclear incompatibilities, which for amphibians have previously been documented to affect hybridisation beyond Haldane's rule at the between-species level.

Reviewer #2 (Remarks to the Author):

In this manuscript, the authors analyse the effects of crossing several populations of the frog *Pseudophryne guentheri*, which is affected by habitat fragmentation and population declines in Australia. The authors crossed females from three and males from four populations and measured fitness related traits in the tadpoles. Overall they found that crosses among genetically and geographic more distant populations resulted in outbreeding (low larval survival and malformations) while crosses between more closely related populations resulted in hybrid vigour (e.g. higher tolerance against desiccation) or had neutral effects. Overall, the manuscript is well written, the experimental design and statistical analyses appropriate. I find the findings of this manuscript of utmost importance for the management of endangered species. Particularly in the face of climate change, research on how to improve the adaptability of populations to climatic changes should be of research priority. The results of these studies can be implemented in the management by applying targeted gene flow into populations of conservation concern. In next future, I would like to see much more of these kinds of study, also because they reveal that local

adaptation as well as geographic and genetic distance between populations within the same species play an important role for conservation measures.

I have only a few suggestions for improving the manuscript:

The Title of at least the Abstract should mention the name of the frog species under investigation.

Line 301: change "that" to "than"

Line 460: Not clear, what you mean by "observation level", please add some information

Response to reviewers

Reviewer comment	Response from Authors (line numbers are those visible with tracked changes hidden)
R1-C1: The main drawback of the work is, however, that I somewhat fail to find an overarching take-home message. From a conservation perspective, the experimental findings did not lead to the development of a tangible management plan.	Our study species, the crawling frog (Pseudophryne guentheri) is one of approximately 250 anuran amphibians occurring in Australia, and is among 70% of species not listed as threatened by any jurisdiction. Consequently there are no documents to guide the management of this species, and until recently there was little evidence that the species may be threatened. A study from our group was the first to show via genomic data that populations are likely declining and losing genetic diversity due to habitat fragmentation (Cummins et al. 2019). Our current study testing targeted gene flow (TGF) is in direct response to this finding, and as far as we are aware is the first test of targeted gene flow in an amphibian. While we agree that ultimately it would be desirable to develop a management plan for this species, particularly one that focuses on responding to the threat of climate change, our strongest take home message is that that the outcomes of TGF will be difficult to predict without further experimentation, and may not be generalisable between, or even within species. This point is already made in the Discussion (lines 244-246, 376-8), and is supported by examples of contrasting results for congeneric species when populations have been mixed (lines 206-8, 364-68, lines 378-81 – no changes to original wording). A further important management implication of our findings is that male population effects are important in P. guentheri, and we include a paragraph (lines 335-349, no changes) that highlights that introducing male gametes to populations is logistically feasible and may therefore yield promising conservation outcomes in terms of TGF. Consequently, given that conservation planning for P. guentheri has not commenced, we focused the final part of our discussion of conservation planning for threatened amphibians in Australia – particularly the Northern and Southern corroboree frogs (lines 362-4). These species are Critically Endangered under Australian legislation, and threatened by climate change and disease. We pointed to plans for initiating targeted gene flow to increase resistance to disease in key populations (lines 362-3), and now also highlight that many species are in captive breeding programs and subject to research on gamete harvesting (new lines 368-73) as follows: These and several other threatened amphibian species are subject to captive breeding programs and efforts to improve the capacity to generate, harvest and store gametes⁸⁰⁻⁸¹. Given that application of TGF in these species will require both a reliable source of adults, and a supply of viable eggs and sperm, we are

	well positioned to initiate laboratory trials that target the expression of traits that could enhance survival of wild populations as climates become increasingly stressful.' There is also a further paragraph in our manuscript pointing out that gamete compatibility may affect the ability to employ TGF in amphibians (lines 252-261) In the light of these points, we argue that we do develop strong narratives around conservation planning in our manuscript. However, we see little value in suggesting management strategies for P. guentheri, given that there is more to understand about how TGF could be employed in this species.
R1-C2: From a conceptual perspective, the underlying reasons for the observed fitness responses could not be further investigated, mainly because the study individuals were not genotyped.	We agree with the reviewer that the underlying reasons for fitness responses are not fully resolved, particularly given that the asymmetrical results for one two-way cross raised the possibility that traits expressed in offspring are influenced by mito-nuclear compatibility (i.e. the maternally inherited mitogenome interacting with the recombined nuclear genome of both parents). The reviewer makes a good point that genotyping parents and offspring would have allowed us to partition the genetic variation in traits (as we did via quantitative genetics in a related study – Eads et al. 2012), but we argue that we are now in a better position to design a follow up study, as our results suggest that sequencing nuclear and mitochondrial genes will each be important for understanding how physiological and metabolic traits are expressed (e.g. Lee-Yaw et al. 2014, now cited at Line 232 in response to R1-C6). The details of a design of a follow-up study, however, are beyond the scope of this manuscript.
R1-C3: I found the entire Discussion -as the authors themselves admit at line 288- at times rather speculative, and also a little unfocused throughout.	There is some speculation in our discussion, as we acknowledged at line 335 (formerly line 288), much of it stemming from the unexpected result that the 'direction' of a within-species cross (i.e. whether sperm from one population are mixed with eggs from another, or vice versa) affected whether the outcome of TGF outcome was neutral or positive. Our discussion systematically considers all important components of our results, which are not straightforward, and in each case we offer suggestions for underlying mechanisms. For example, in addition to discussing mitochondrial/nuclear interactions (see response to R1-C2), we also raise the possibility that dosage effects mediated by polyploidy (known in other amphibian species in the south-western Australian bioregion) could produce asymmetrical responses when populations are crossed in different directions (lines 328-31, no changes). We logically then conclude that ploidy should be investigated in this species (lines 331-3, no changes) As our present investigation is built upon recent research on the

	physiology (Rudin-Bitterli et al. 2020b), genetics (Eads et al. 2012, Cummins et al. 2019) and reproductive traits (Rudin-Bitterli et al. 2020b) of this species, we have been able to be much less speculative about the mechanisms, risks and benefits of TGF than if we had initiated this research on a poorly known species. For example, thanks to these earlier studies, we are able to report both the geographic and genetic distances between the populations we crossed, and we have detailed knowledge of ejaculate traits that have helped identify a possible mechanism for outbreeding depression. As such, it would be difficult to write a discussion for any amphibian species subject to tests of TGF that would not have many areas of uncertainty.
R1-C4: The manuscript could become vastly strengthened for example if the findings from the experimental crosses are underpinned by individual-specific genetic data – I wonder if the SNP genotypes published in a previous paper by the authors were based on the study individuals of the present manuscript?	We agree that ultimately the questions raised by our results – particularly regarding asymmetry in trait responses in the crosses of populations A and B - could be resolved with the addition of genetic data. We are pleased that Reviewer 1 is familiar with our study system, as they mention research (Cummins et al. 2019) that used genomic markers (SNPs) to explore patterns of local adaptation related to environmental gradients (temperature and rainfall). Genetic evidence for local adaptation was further supported by a physiological study (conducted earlier but published later – Rudin-Bitterli et al. 2020a) that showed that adults and embryonic stages were adapted to different hydric environments across an environmental gradient. The adults and some of their offspring in the physiological study (Rudin-Bitterli et al. 2020a) were the same 192 individuals analysed in the genomic study (Cummins et al. 2019), which may explain Reviewer 1’s expectation of genetic data in the current study. However, in this study on the outcomes of mixing xeric and mesic populations, we do not have genetic data for the parent frogs, nor do we have genetic data for the almost 3000 embryos and tadpoles created via experimental crosses. All tadpoles for which we measured a variety of traits shortly after hatching were preserved in buffered formalin, which minimises shrinkage of soft tissues, and so leads to reliable morphological measurements. Unfortunately this prevented subsequent sampling of tadpole tissue for DNA extraction, as high quality DNA is currently difficult to extract from material fixed in formalin. Ultimately, and with the benefit of hindsight, an experiment could be designed that would allow measurements of traits and genotyping (e.g. tadpoles or metamorphs could be anaesthetised for mass and morphological measurements before being preserved in a buffer suitable for DNA extraction). This is an approach we are likely to adopt in a future study.
R1-C5: the parentage of the	Given that we do not have any genetic data (which again,

hatchling cohorts could be determined (the mixing of sperm from 5 males leaves between 1 and 5 possible sires), to reveal possible effects of population crosses on levels of sperm competition.	Reviewer 1 might have expected based on their knowledge of Cummins et al.), we cannot investigate sperm competition. While undoubtedly this is a relevant phenomenon in this species based on our recent study of ejaculate traits (Rudin-Bitterli et al. 2020b), adding such an analysis might distract from our focus on TGF. As we explain at lines 430-3 (unchanged), the reason we used sperm suspensions from multiple males was to reduce the impact of individual-specific pairwise interactions that might override any population-level effects we were interested in.
R1-C6: As a further possible inference, mitochondrial genotypes could be linked to the outcome of crosses, particularly in light of the finding that sex-reciprocal crosses between given population pairs lead to differential phenotypic fitness.	This is an excellent suggestion, and note that it reflects our discussion point that mitotypes might be responsible for different phenotypes arising from crosses of the same two populations in different directions (lines 323-5, unchanged). However, given that we have no genetic data (nuclear or mitochondrial) to draw upon, we are not able to speculate further.
R1-C7: At present, the authors largely neglect the possible presence of cytonuclear incompatibilities, which for amphibians have previously been documented to affect hybridisation beyond Haldane's rule at the between-species level.	The reviewer is correct to point out that we neglected to specifically mention studies of cytonuclear incompatibility (aka mitonuclear incompatibility) in amphibians, and we have subsequently identified two examples relevant to our work. Consequently, we have made changes in two parts of the Discussion. The first is in a section on Asymmetric genetic incompatibilities, where the new text reads (lines 229-232): 'Additionally, cytonuclear incompatibility (=discordance) is likely to explain asymmetrical outcomes in reciprocal crosses of two species of newt (Triturus cristatus and T. marmoratus)⁴⁵, and differential growth performance across mitotypes of the salamanders Ambystoma macrodactylum⁴⁶' The second instance is at lines 326-7, where we have adjusted a sentence that was inaccurate (i.e. we now cite the single genetic study of mitotypes in amphibians and their relation to offspring traits): Mitonuclear/environmental interactions have recently been shown in insects⁷² and birds⁷³, but mitonuclear co-evolution has received limited attention in amphibian populations⁴⁶
R2: The Title or (sic) at least the Abstract should mention the name of the frog species under investigation.	The species common name and scientific name are now included in the Abstract. These details were originally omitted due to the restricted word count of 150, and the word count is now 156. This could be reduced if the common name is omitted. Line 18-19 now reads: ' We reared embryos of the crawling frog (Pseudophryne guentheri) on wet and dry soils...'

R2: Line 301: change "that" to "than"	Thank you for picking up this typo. Now corrected (Line 348)
R2: Line 460: Not clear, what you mean by "observation level", please add some information.	This refers to a simple statistical 'fix' for over-dispersed data (often used in binomial models). The observation level is where each data point receives a unique level of random effect (i.e. for each individual) and including this term models the overdispersion in our data. We thank the review for seeking clarification here, as we note we did not include the reference (Harrison 2014) in the appropriate format. This has now been corrected, as below. Lines 519-21 now read: 'Only one trait, embryonic survival, was overdispersed, which if uncontrolled can lead to biased parameter estimates. To account for this overdispersion, we included an observation-level random effect (where each data point in our analysis receives a unique level of random effect) when analyzing this trait⁹⁸.'

REVIEWERS' COMMENTS:

Reviewer #2 (Remarks to the Author):

This is the second time I look at this manuscript. All issues raised by the reviewers have become addressed in detail, with convincing arguments provided in the rebuttal letter also when specific suggestions could not be taken into account (such as analyses based on genotypes for the study individuals). As a whole, the manuscript itself only became rather marginally amended in the course of the review process, with however useful additions to the Discussion to widen its scope.

All papers have their limitations, and overall this is a very fine piece of work. I only found very minor issues which can probably be dealt with at the page proof stage.

Line 292-293: 'For example, there are numerous examples.....' reads a little awkward. Delete the first 'example'?

Table 1 and its legend: be consistent with italicizing (in e.g. 'Fis' or 'He', the subscript should not be italicized, but the main letter should be).

Reviewer #3 (Remarks to the Author):

Excellent and urgently needed study!
Can be accepted for publication.